# Attention-Deficit/Hyperactivity Disorder Symptoms, Sensation-Seeking, and Sensory Modulation Dysfunction in Substance Use Disorder: A Cross Sectional Two-Group Comparative Study

**DOI:** 10.3390/ijerph19052541

**Published:** 2022-02-22

**Authors:** Naama Assayag, Itai Berger, Shula Parush, Haim Mell, Tami Bar-Shalita

**Affiliations:** 1School of Occupational Therapy, Faculty of Medicine, Hebrew University of Jerusalem, Jerusalem 9112102, Israel; naama.lasri@mail.huji.ac.il (N.A.); shula.parush@mail.huji.ac.il (S.P.); 2Pediatric Neurology, Pediatric Division, Assuta Ashdod University Hospital, Faculty of Health Sciences, Ben-Gurion University, Beer-Sheva 8443944, Israel; dr.itai.berger@gmail.com; 3School of Social Work and Social Welfare, Hebrew University of Jerusalem, Jerusalem 9190501, Israel; 4Department of Criminology, Max Stern Yezreel Valley College, Yezreel Valley 1930600, Israel; mellhaim@012.net.il; 5Department of Occupational Therapy, School of Health Professions, Faculty of Medicine, Tel Aviv University, Tel Aviv 6997801, Israel

**Keywords:** SUD, ADHD, sensation-seeking, sensory processing, sensory over-responsiveness, risk factor, therapeutic community

## Abstract

Background: Attention-deficit/hyperactivity disorder (ADHD) and sensation-seeking, a trait characterized by risk-related behaviors, have been recognized as risk factors in substance use disorder (SUD). Though ADHD co-occurs with sensory modulation dysfunction (SMD), SMD has scarcely been explored in SUD. Thus, this study aimed to characterize ADHD symptomology, sensation-seeking, and SMD, as well as to explore their contribution to SUD likelihood. Methods: A cross sectional two-group comparative study including therapeutic community residents with SUD (*n* = 58; study group) and healthy individuals (*n* = 62; comparison group) applying the MOXO continuous performance test (MOXO-CPT) evaluating ADHD-related symptoms. In addition, participants completed the ADHD Self-Report Scale—Version 1.1 for ADHD screening; the Brief Sensation Seeking Scale quantifying risk-taking behaviors; and the Sensory Responsiveness Questionnaire-Intensity Scale for identifying SMD. Results: The study group demonstrated higher SMD incidence (53.57% vs. 14.52%) and lower performance in three MOXO-CPT indexes: Attention, Impulsivity, and Hyperactivity, but not in Timing, compared to the comparison group. Sensory over-responsiveness had the strongest relationship with SUD, indicating 27-times increased odds for SUD (95% CI = 5.965, 121.216; *p* ≤ 0.0001). A probability risk index is proposed. Conclusion: We found SMD with the strongest relation to SUD exceeding that of ADHD, thus contributing a new perspective for developing future therapeutic modalities. Our findings highlight the need to address SMD above and beyond ADHD symptomology throughout the SUD rehabilitation.

## 1. Introduction

The bidirectional link of substance use disorder (SUD) and attention-deficit/hyperactivity disorder (ADHD) is widely described [1,2,3,4], although its nature remains unclear [5]. ADHD, a neurodevelopmental disorder characterized by inattentive or hyperactive/impulsive behavior, or both [6], is associated with impairment in multiple life domains [7,8,9]. A meta-analysis found that 23.1% of those seeking SUD treatment have ADHD [10]. Moreover, ADHD has been reported as a risk factor for SUD [11,12]. Interestingly, ADHD is related to traits such as sensation-seeking [13], included under the broader umbrella of disinhibition [14], characterized by the desire for intense and novel experiences [15], and considered a trait of risk-taking behavior [16]. Sensation-seeking has consistently been related to higher incidences of alcohol and substance abuse [17,18], recognized as a risk factor for adolescent substance use [14,18], and for the development and maintenance of SUD. ADHD has also been found as an independent predictor of sensation-seeking [19].

Recent years have provided a sound base of evidence regarding the co-occurrence of ADHD and sensory modulation dysfunction (SMD) [20,21,22,23]. SMD is a neurodevelopmental sensory-processing alteration, characterized by difficulty in regulating the degree, nature, or intensity of responses to sensory stimulation in single or multiple sensory systems [24,25,26]. Its clinical manifestations are characterized by sensory under-responsivity (SMD-SUR), demonstrated by disregarded or delayed responses to stimulation; and sensory over-responsivity (SMD-SOR), perceiving non-painful sensations as irritating, unpleasant [25,27], or painful [28,29,30]. The sensory realm has been neglected over the years [31], and only recently, initial findings indicate SMD in SUD [32,33,34,35,36]. Importantly, our recent work found that 54% of individuals with SUD were also identified as having SMD; specifically, 47% were identified with SMD-SOR [32]. Evidence supports ADHD and SMD as distinct conditions, e.g., [20,21,37], yet the differential diagnosis between SMD and ADHD is often challenging, due to the overlapping symptoms and high rate of co-morbidities [37].

As far as we know, the co-occurrence of ADHD and SMD has not yet been explored among individuals with SUD, leaving the contribution of SMD, as well as its distinct profile in the SUD phenomenon, empirically sparse. Further, since SMD severely interferes with participation in everyday activities [38,39], and impacts quality of life [28,40], exploring the contribution of SMD, beyond and above ADHD and sensation-seeking symptomology, can deepen the understanding of the risks and trajectories for one of the most global health concerns, and, in the future, may add a new therapeutic modality for individuals with SUD.

Therefore, the aims of this study were to characterize attention-deficit/hyperactivity disorder symptomology, sensation-seeking, and SMD among therapeutic community residents with SUD, and to examine each contribution to SUD likelihood.

## 2. Materials and Methods

This research has been approved by the Institutional Ethical Review Committee Board (IRB), Tel Aviv University (11002976_20160720). All participants were informed of the research objectives and possible inconveniences, and signed an informed consent.

### 2.1. Study Design

This was a cross sectional two-group comparative study.

### 2.2. Participants

The study group comprised individuals aged 18–54 years, residing in a therapeutic community (TC) in northern Israel. All of the TC residents meet the diagnostic criteria for severe SUD according to the Diagnostic and Statistical Manual of Mental Disorders-IV (DSM-IV) [6], and had no dual diagnosis and no cognitive deficits. The TC is a rehabilitation center providing a controlled drug-free environment with multidimensional support [41,42], lasting typically 1 to 1.5 years. Abstinence is routinely verified through random urine testing. The comparison group, age-matched, was composed of recruited healthy volunteers from the general population, using the snow-ball sampling method.

For the study group, inclusion criteria stipulated no current use of medication for psychological or neurological disorders; no known brain lesions; adequate language skills; and abstinence from drugs and alcohol for at least 14 days. All individuals in the TC meeting the inclusion criteria participated in the study, comprising the study group. This purposive sample method [43] effectively negated the potential for volunteer bias.

For the comparison group, inclusion criteria were current or past drug and alcohol abuse; past or present neurological, neurodevelopmental, or psychiatric diagnosis, including ADHD according to self-reporting. 

Sample size was based on a study which compared sensory modulation types between people with SUD and healthy people [33]. Calculation was based on power analyses derived from a *p* value of 0.05 and a statistical power of 0.80, yielding *n* = 58 in each group.

### 2.3. Assessments

#### 2.3.1. Performance Testing

The MOXO Continuous Performance Test (MOXO-CPT) [44] was utilized: a standardized, computerized 18.2-min test to diagnose ADHD-related symptoms [45], using target and non-target card images comprising eight blocks (136.5 s, 59 trials each). 

In each trial, the target is presented in the middle of a computer screen for 500, 1000, or 3000 ms, followed by a ‘void’ period of the same duration. Distractor onset is not synchronized with target onset, and could be presented during the void period as well.

Distractors are presented for 8 s, with a fixed interval of 0.5 s between two distractors. Participants are requested to respond to the target stimuli as fast as possible by pressing the space bar only once, and avoid responding to non-target stimuli. Three types of distractions are presented: (a) visual distractors (e.g., animated barking dog); (b) auditory distractors (e.g., barking sound); and (c) a combination of both (e.g., animated barking dog with the sound of barking). Visual distractors appear at one of four spatial locations on the sides of the screen: down, up, left, or right. Overall, eight different distractors were included, and each of them could appear as purely visual, purely auditory, or as a combination of them. The MOXO-CPT provides four indexes: (a) Attention—the number of correct responses to targets not bound by any time frame (max. score 272); (b) Timing—the number of correct responses only while targets are presented (max. score 272); (c) Impulsivity—the number of impulsive responses to non-target stimuli; and (d) Hyperactivity—the remaining commission errors not counted as impulsivity (for example, multiple spacebar presses or random key pressing) (Figure 1).

#### 2.3.2. Self-Report Questionnaires

The Adult ADHD Self-Report Scale-Version 1.1 (ASRS-V1.1) [46] was utilized: a standardized, reliable, and valid 18-item checklist for evaluating adult ADHD based on the DSM-IV diagnostic criteria. Our study utilized the shortened 6-item version on a 5-point Likert scale (‘Never’ (0) to ‘Very Often’ (4)), which reported high agreement with the clinical classification of adult ADHD [47], and has proven beneficial for screening the SUD population [48]. Using the recent adaptation for the ASRS scoring based on Ustun et al. (2017) [49], we applied the score of 11 as the cut-off point. A high internal consistency (Cronbach’s alpha 0.88 [50]), and good test-retest reliability were reported [51]. The 6-item version has a moderate sensitivity of 68.7% and a high specificity of 99.5% [46]. In this study, the ASRS was completed only by the SUD group.

The Brief Sensation Seeking Scale (BSSS) [17] was utilized: a standardized, reliable, and valid 8-item self-report scale measuring sensation-seeking through four dimensions: experience-seeking; boredom susceptibility; thrill- and adventure-seeking; and disinhibition. Responses are indicated on a 5-point Likert scale from ‘strongly disagree’ (1) to ‘strongly agree’ (5). The BSSS has high internal consistency (Cronbach’s alpha 0.76), and solid psychometric characteristics with stability across age, gender, and ethnic categories. 

The Sensory Responsiveness Questionnaire-Intensity Scale (SRQ-IS) [52] was utilized: a standardized, reliable, and valid 58-item scale for identifying SMD. It presents routine activities involving one sensory stimulus in one modality (auditory, visual, gustatory, olfactory, vestibular, and somatosensory, excluding pain). Participants are requested to rate their response on a 5-point Likert scale from ‘not at all’ (1) to ‘very much’ (5). The SRQ provides two scores for each of the two SMD subtypes: SMD-SOR is determined by applying the SRQ-Aversive subscale score for scores higher than the normal cut-off score (mean + 2SD; 1.87 + 0.52); the SMD-SUR subtype is determined by applying the SRQ-Hedonic subscale score for scores higher than the normal cut-off score (mean + 2SD; 2.10 + 0.66). High internal consistency (Cronbach α = 0.90–0.93), test–retest reliability (r = 0.71–0.84, *p* < 0.01–0.05), content, criterion, and construct validity were reported [52].

### 2.4. Procedures

Both groups completed the questionnaires and the MOXO-CPT during one-hour sessions. All data were collected by one researcher (N.A.) in an air-conditioned (22–24 °C) room, with ambient noise typically not exceeding 45dBSPL. 

### 2.5. Data Analysis

Statistical analyses were performed with SAS V9.4 (SAS Institute, Cary, NC, USA). Descriptive statistics (mean, standard deviation, minimum, median and maximum for continuous variables, and a count and percentage for discrete variables) are presented for the study parameters by research group (study/comparison). Categorical variables were compared between the two groups using a chi-squared test, and the continuous variables with a two-sample *t*-test. 

SUD was modeled using logistic regression, and SMD-SOR/non-SMD-SOR status, SMD-SUR/non-SMD-SUR status, MOXO-CPT four indexes, and BSSS-Total score were entered as potential risk factors. Odds ratios (95% confidence intervals) were presented, and significant variables (*p* < 0.05) were entered into a multivariate model. The variables that remained in the model were those which remained statistically significant when entered together, and maximized the predictive power (area under the curve (AUC) of the receiver operating characteristic (ROC) curve) of the model, such that the AUC of the resulting ROC curve was at least 0.8. The risk score, which was calculated from a linear combination of the logistic regression model coefficients, is presented in an effect plot portraying the risk of having SUD as a function of the SMD sub-type, MOXO-CPT indexes, and BSSS-Total score. All statistical tests were two-sided. A *p*-value of 0.05 or lower was considered statistically significant. No adjustments for multiple testing were performed, as this is a report of a preliminary examination of these associations; thus, only nominal *p*-values are presented.

## 3. Results

### 3.1. Demographic Characteristics

One hundred and twenty individuals (89% men (*n* = 107)), mean (SD) age 26.7 (8.08), divided into the study (SUD, *n* = 58) and comparison (healthy, *n* = 62) groups, participated in this study. Within the study group, 33% of participants screened positive for ADHD. No statistically significant group differences were found with respect to sex (χ^2^ (1, *N* = 120) = 0.57, *p* = 0.4507) or age (*t*(91) = 0.89, *p* = 0.3776). Formal education (years) was found to be significantly different (*t*(98) = −7.35, *p* < 0.0001), revealing fewer years of education in the study group (Table 1).

### 3.2. Substance Use Consumption in the Study Group

Substances were divided into five categories [53] based on self-reported preferred psychoactive substance: Cannabis, Opioids, Stimulants, Synthetic Cannabinoid, and Alcohol. The most-reported past consumed substance in the study group was Synthetic Cannabinoid, whereas the least consumed were Stimulants (Table 1).

### 3.3. Group Differences in the MOXO-CPT Indexes

Significant group differences were found in the three MOXO-CPT indexes: Attention, Impulsivity, and Hyperactivity (*t*(97) = −3.42, *p* = 0.0009; *t*(97) = 3.83, *p* = 0.0002; *t*(97) = 3.81, *p* = 0.0002), but not in Timing (Table 2).

### 3.4. Group Differences in the BSSS Dimensions

Statistically significant group differences were found in the BSSS-Total score and in the three BSSS dimensions: Boredom Susceptibility; Thrill- and Adventure-Seeking; and Disinhibition ((*t*(110) = 5.23, *p* < 0.0001); *t*(110) = 3.61, *p* = 0.0005; *t*(110) = 4.42, *p* < 0.0001; *t*(110) = 5.97, *p* < 0.0001, respectively) (Table 2).

### 3.5. SMD Distribution in the Study and Comparison Groups

A statistically significant group difference was found in the SMD incidence, demonstrating a higher rate in the study group (53.57%) comprising mostly SMD-SOR vs. the comparison group (14.52%) (SMD: χ^2^(1) = 20.28, *p* < 0.0001; SMD-SOR: χ^2^(1) = 23.99, *p* < 0.0001) (Table 2). No statistically significant group difference was found in SMD-SUR incidence (χ^2^(1) = 1.98; *p* = 0.16) (Table 2). Statistically significant group differences were also found in both SRQ scores, showing elevated scores in the study group: (SRQ-Aversive mean (SD): study group: 2.3 (0.52) vs. comparison group: 1.9 (0.30), *t*(116) = 5.15, *p* < 0.0001; and SRQ-Hedonic mean (SD): study group: 2.3 (0.43) vs. comparison group: 2.06 (0.36), *t*(116) = 3.53, *p* = 0.0006).

### 3.6. SMD, MOXO-CPT, and Sensation-Seeking as Risk Factors for SUD

SMD-SOR (Yes/No), MOXO-CPT Impulsivity Index, and BSSS-Total score were found to be significantly related to SUD in a multivariate model and in ROC analysis, by maximizing the area under the curve (AUC). The additional variables of the MOXO-CPT Impulsivity Index and the BSSS-Total score significantly improved the AUC of the test, reaching 0.904 (95% Wald CI: (0.8431–0.9658)) versus the model with SMD-SOR alone (AUC = 0.7281; 95% Wald CI: (0.6409–0.8152)). Table 3 presents the adjusted odds ratios, 95% confidence intervals, and levels of significance. SMD-SOR was found to be the strongest risk factor for SUD: subjects with SMD-SOR were found to be at 26.889-times higher risk than subjects without SMD-SOR (Table 3). 

The logistic regression model coefficients were used to derive a probability index as *P = eY*/*(1 + eY),* where *Y* is a linear combination of the model coefficients. The probability index is presented on a scale of 0 to 1, where a higher score indicates a higher likelihood for having SUD. Figure 2 demonstrates the predicted SUD probability score value for individuals with and without SMD-SOR, based on their BSSS-Total score and MOXO-CPT Impulsivity Index. Whereas the probability index score for SUD for a person without SMD-SOR with a BSSS Total score of 3 and a MOXO-CPT Impulsivity score of 20 is about 0.20 (20%), the probability for someone with the same scores who also has SMD-SOR escalates to nearly 0.80 (80%).

## 4. Discussion

The aims of this study were to characterize attention-deficit/hyperactivity disorder symptomology, sensation-seeking, and sensory modulation dysfunction among TC residents with substance use disorder, and to examine each contribution to substance use disorder likelihood. Although each of the variables separately was previously reported in the SUD population [19,32,54], to the best of our knowledge, this is the first study to examine the interplay between ADHD symptomology, sensation-seeking and SMD in individuals with SUD residing in a TC. The study group displayed higher incidence of SMD- and ADHD-related symptoms compared to the comparison group. Though ADHD is a known risk factor for SUD, our data suggests that SMD-SOR has a higher incidence and a stronger link to SUD compared to ADHD. 

This study found an ADHD prevalence of 33%, closely approximating ADHD in SUD-reported prevalence [48]. Though most studies examined ADHD symptoms as a group or according to the diagnosis subtypes (mainly inattentive, mainly hyperactive/impulsive, or combined type) [55,56], this current study quantified each ADHD symptom independently using the MOXO-CPT (e.g., Attention, Timing, Impulsivity, and Hyperactivity), and found that the SUD group demonstrated lower performance in three MOXO-CPT indexes: Attention, Impulsivity, and Hyperactivity. Our results are in line with previous research showing that the MOXO-CPT distinguished healthy controls from individuals with ADHD, individuals with SUD, and individuals with comorbid ADHD and SUD [57]. Since no significant group difference was found in Timing, which differentiates between motor-speed difficulties and inattention [58], we propose that the Attention performance is the one responsible for the disparity between the groups. These results lend further support to the notion that most CPT tasks require multiple cognitive abilities [58]. Therefore, an integration of CPT indexes may better reflect the complexity and heterogeneity of ADHD etiology and clinical manifestations, especially when combined with complex comorbidities. Specifically, using the MOXO-CPT, we found that the impulsivity surpassed the other indexes, and was the most potent risk factor for SUD. This finding supports studies suggesting that impulsivity is an important trait of risk behavior, and plays a major role in SUD [59,60,61] and SUD vulnerability. However, it should be taken into account that ADHD and SUD have complicated and bi-directional relationships compounding and maintaining the symptoms of each other [62]. Due to this relationship, ADHD symptoms can not only predispose SUD, but can be adversely be affected by SUD [4]. Thus, the cross-sectional design in this study impedes any conclusions concerning the causal relationship.

Indeed, in addition to ADHD, our study looked at sensation-seeking behavior as a trait of risk-taking behavior [16]. This study validates previous findings of higher sensation-seeking behavior in individuals with SUD compared to healthy individuals [18]. From a sociological perspective, sensation-seeking is an individual interpersonal trait that is the result of reciprocal and reinforcing social influences, implying that the social environment has the ability to limit the negative outcomes of sensation-seeking behavior [63]. Neurophysiologically, sensation-seeking operates under the same neural structures involved in the reward system [64]. Thus, the risky nature of SUD, which meets the desire for sensation-seeking by providing the necessary stimulation [17], may indicate that the sensation-seeking profile we found is a SUD consequence.

Interestingly, Yalachkov, Kasiser, and Naumer (2010) [31] proposed a model describing the way that sensory processing might be involved in addiction mechanisms by stimulating the reward system. According to this model, cue exposure can elicit activations of the sensory and motor representations, which, in turn, activate the reward system, and contribute to an increased likelihood of relapse. As individuals with SMD are characterized by alertness to environmental stimuli, as evidenced by their EEG pattern of response [65], it could be suggested that, in individuals with SMD, a stronger response to environmental stimuli of substance-related cues may be generated. Although the mechanisms governing SMD and the reward systems are yet inconclusive, Borges et al. (2017) found that sensory imbalance was implicated in a decreased resilience to psychoactive substance use. Of note, though balanced sensory processing improves our ability to respond, learn, detect, discriminate, and recognize information from the environment [66], SMD is a product of sensory imbalance which can lead to difficulties managing maladaptive behaviors [67]. Because of its importance in everyday functioning, it is possible to assume that a dysfunctional sensory system could impact daily information processing [25], and therefore have a detrimental effect on a wide variety of health conditions, including SUD. Importantly, the role of the sensory system has been neglected in SUD research [31], leaving a gap in knowledge. 

A different perspective suggests the self-medication hypothesis (SMH) [68] explaining the high incidence of SMD in SUD [32]. Namely, a psychoactive substance is used as a coping mechanism for minimizing or avoiding emotional suffering [68,69,70]. Specifically, since the SMD manifestations include pain [71,72] anxiety [40,73,74], psychological distress [75], negative affect [76], and depression [40], it could be proposed that individuals with SMD choose substance use as a coping mechanism [32]. However, although the relationship between SMD and SUD is scarcely reported, the SMD profile we found may be a SUD consequence, and not a predisposition.

Importantly, finding SMD with the strongest association to SUD is worthy both clinically and for research. Although, to date, ADHD and sensation-seeking have been well established in the literature as risk factors for SUD [12], we found SMD-SOR to be a stronger risk factor than both of these. The probability risk index this study proposes indicates that for identical scores in the sensation-seeking and MOXO-CPT-Impulsivity measures, the presence of SMD-SOR significantly increases the likelihood of SUD. To our knowledge, this is a novel approach not previously explored in SUD. These findings are in line with previous reports suggesting SMD as a predisposing factor serving as a risk factor for other health conditions [77,78]. Our findings contribute significantly for the better understanding of SUD, as well as suggesting a new perspective for both prevention and rehabilitation as part of the broader SUD treatment. 

### Limitations

Some limitations should be considered, including data collection from a single TC, which could limit the generalization of the results. The unequal sex distribution, though similar to other studies [79], may interfere with generalizability to the female population. In addition, no standard cognitive assessment was performed. Finally, most of the outcome measures were based on self-reporting, which may be subject to response biases. Future studies require using a larger sample size consisting of various TCs with a larger female cohort, such that sensory modulation sub-types could be studied in relation to the preferred psychoactive substance consumed.

## 5. Conclusions

Our results emphasize that SMD has the strongest relation to SUD, exceeding that of ADHD. Moreover, our study emphasizes that SMD significantly increases the likelihood of SUD. These findings highlight the need to address sensory modulation in the course of SUD rehabilitation, in addition to attention, impulsivity, and hyperactivity. Therefore, our findings may suggest a new perspective for both prevention and rehabilitation as part of the broader SUD treatment.

## Figures and Tables

**Figure 1 ijerph-19-02541-f001:**
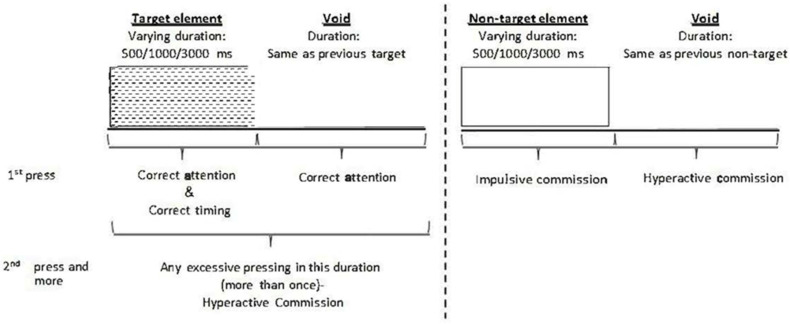
MOXO-CPT description. MOXO-CPT, MOXO continuous performance test; ms, milliseconds. Definition of the timeline, target, and non-target stimuli were presented for 500, 1000, or 3000 ms. Each stimulus was followed by a void period of the same duration. The stimulus remained on screen for the full duration regardless of the response. Distracting stimuli were not synchronized with target/non-target onset, and could be generated during target/non target stimulus or during the void period.

**Figure 2 ijerph-19-02541-f002:**
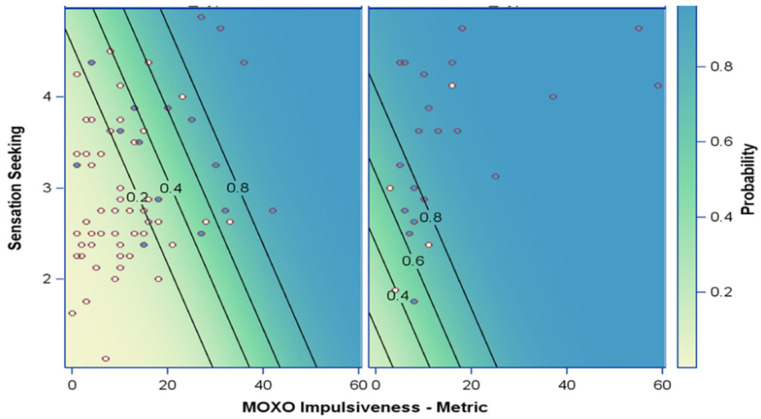
Risk index for SUD. The probability index for SUD takes into account the BSSS-Total score and the MOXO-CPT Impulsivity Index, differentiating between persons with and without SMD-SOR. SMD-SOR, sensory over-responsivity; MOXO-CPT, MOXO continuous performance test; SUD, substance use disorder.

**Table 1 ijerph-19-02541-t001:** Gender, age, and years of education distribution in the two groups, and substance use divided into categories in the study group.

Charactristics	Study Group *n* = 58	Comparison Group *n* = 62
Gender		
Male	91.4% (*n* = 53)	87.1% (*n* = 54)
Female	8.6% (*n* = 5)	12.9% (*n* = 8)
Age (years), Mean (SD)	27.4 (9.94)	26.0 (5.83)
Years of education, Mean (SD)	11.4 (1.30)	13.9 (2.27)
Substance use distribution *		
Cannabis	22.4% (*n* = 13)	
Opioids	22.4% (*n* = 13)	
Stimulants	8.6% (*n* = 5)	
Synthetic cannabinoid	31% (*n* = 18)	
Alcohol	15.5% (*n* = 9)	
Initial age for drug use, Mean (SD)	16.4 (3.78)	

* Substance use distribution according to preferred psychoactive substance in the study group. SD, standard deviation.

**Table 2 ijerph-19-02541-t002:** Distribution of SMD (% (*n*)), BSSS, and MOXO-CPT indexes (mean (SD)) in both groups.

Study Variables	Study Group (*n* = 58)	Comparison Group (*n* = 62)
SMD	53.6 (30)	14.5 (9)
SMD-SUR	14.3 (8)	6.5 (4)
SMD-SOR	48.2 (27)	8/1 (5)
BSSS-Total score	3.5 (0.73)	2.8 (0.75)
Experience-Seeking	3.6 (1.00)	3.4 (1.03)
Boredom Susceptibility	3.1 (0.74)	2.5 (0.90)
Thrill- and Adventure-Seeking	3.6 (1.19)	2.7 (0.98)
MOXO-CPT *		
Disinhibition	3.7(1.09)	2.5 (1.09)
Attention	265.1 (9.55)	269.6 (2.61)
Timing	232.9 (25.27)	233.4 (24.31)
Hyperactivity	9.4 (11.96)	3.2 (2.79)
Impulsivity	17.7 (13.53)	9.7 (6.95)

SMD, sensory modulation dysfunction; SMD-SUR, sensory under-responsivity; SMD-SOR, sensory over-responsivity; BSSS, Brief Sensation Seeking Scale; SD, standard deviations (higher scores (range 1–5) denote higher levels of sensation-seeking); MOXO-CPT, MOXO continuous performance test; for Attention and Timing indexes (max. score 272), higher scores denote better performance. For Hyperactivity and Impulsivity indexes (min. score 0), lower scores denote better performance; * Moxo *n* in the study group = 42.

**Table 3 ijerph-19-02541-t003:** The likelihood for SUD by Sensory Over-Responsivity, MOXO-CPT Impulsivity, and Sensation-Seeking.

Variable	OR	Lower 95% CI	Upper 95% CI	Pr > ChiSq
SMD-SOR (Yes/No)	26.889	5.965	121.216	<0.0001
MOXO-CPT Impulsivity Index	1.136	1.056	1.222	0.0006
BSSS-Total score	2.877	2.877	6.455	0.0103

SUD, substance use disorder; OR, odds ratio; 95% CI, 95% confidence interval; SMD-SOR, sensory over-responsivity; SMD-SUR, sensory under-responsivity; MOXO-CPT, MOXO continuous performance test; BSSS, Brief Sensation Seeking Scale.

## Data Availability

The data that support the findings of this study are available from the corresponding author, upon reasonable request.

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
