# Peer review of "Attention-Deficit/Hyperactivity Disorder Symptoms, Sensation-Seeking, and Sensory Modulation Dysfunction in Substance Use Disorder: A Cross Sectional Two-Group Comparative Study"

_ijerph, 2022, doi:10.3390/ijerph19052541_

Round 1
Reviewer 1 Report
Congratulations for your work and effort. However, I would like you to pay attention to the indications attached for improvement its importance. 
Paper Title: ADHD, sensation seeking, and sensory modulation dysfunc-2 tion in substance use disorder: Proof of concept study
Journal Manuscript ID: IJERPH-1588136
REVIEW
This study wants to describe attention-deficit/hyperactivity disorder symptomology, risk-related behaviors, and sensory modulation dysfunction among therapeutic community residents with substance use disorder.
I think that the focus of this study could be important. Congratulations for your work and effort. However, I would like you to pay attention to the following indications for improvement its importance. 
Abstract:
- Title: It is strongly recommended that in the title you add information on the type of study design. Currently it is indicated "Proof of concept Study", but in the text, in the Methods section, it is indicated that it is a "Cross sectional two-group comparative study". It is suggested that the title should state the same.
- Background: Please, remove the word “Objetive”, since you are providing a background or information that describes why this study is being conducted.
- Methods: Please, include the study design.
- Conclusions: Please clarify how the findings in the study could affect intervention.
Introduction:
Based on the information presented in the introduction, key information missing pertains to the following:
- The introduction is very weak. I need the following questions (that make up the introduction) to be answered in more depth: What is the problem? Why is it important? What is known so far (scientific articles)? What is not known? Link to the objectives of the study.
- It is said in line 58: “However, the co-occurrence of ADHD and SMD have not yet been tested among individuals with SUD.” Why?? Why has it not been studied so far? Please, clarify it.
Methodology
Information provided in the methods and procedures is not detailed or clear enough to replicate the study. See below for suggestions/ feedback.
- The setting and location of the study are unclear. Based on information presented, the setting may have occurred in a therapeutic community. How long have the residents been there? What is their condition? Close to the discharge? Based on the discussion of the study, the location may have occurred in Israel. Is that right? Please, include this information in the methods section.
- Where healthy volunteers are obtained? Are they searched by age matching?
- The process of participant selection and eligibility criteria are quite confused. Please, clarify and rewrite the inclusion and exclusion criteria section. Do not jump from one to another.
- Line 93, there is a gap.
- What is the cognitive level? Is the level of cognitive reserve known?
- It is missing sample size calculation.
- It would be interesting to know which type of consumption is more related to sensory seeking or sensory modulation. Why don't you consider this analysis?
Results
- Table 1, it is missing a parenthesis “Initial age for drug use, Mean (SD”
- Table 1 is very simple, it would be more interesting to know data such as: Education (compulsory education, vocational college…), Employement, Marital status… If they have consumed medication for hyperactivity, Nationality, etc.
- Table 2, “n" is equal to 56, but 58 is indicated above. Is there is missing data?
- It would be interesting to know if depending on the drug consumed there is more sensory search.
Discusión
- It would be good starting with the purpose of the study.
- It is said at the first pharagrahp that “…our study demostred that…”. Please, be careful with your words, because the sample may not be large enough to extrapolate the results. I would use sentences such as: “the data suggest....” “It seems to…”.
- It is said, line 243: “…this is the first study to examine…”. Perhaps the statement is a bit pretentious, since there are similar studies, although not the same, such as:
Assayag N, Bonneh Y, Parush S, Mell H, Kaplan Neeman R, Bar-Shalita T. Perceived Sensitivity to Pain and Responsiveness to Non-noxious Sensation in Substance Use Disorder. Pain Med. 2020 Sep 1;21(9):1902-1912. doi: 10.1093/pm/pnz292.
Moggi F, Schorno D, Soravia LM, Mohler-Kuo M, Estévez-Lamorte N, Studer J, Gmel G. Screened Attention Deficit/Hyperactivity Disorder as a Predictor of Substance Use Initiation and Escalation in Early Adulthood and the Role of Self-Reported Conduct Disorder and Sensation Seeking: A 5-Year Longitudinal Study with Young Adult Swiss Men. Eur Addict Res. 2020;26(4-5):233-244. doi: 10.1159/000508304.
In any case, please explain why you believe that no one has studied so far these variables as a whole.
- The limitations of the study are not adequately discussed. Also consider biases related to participant selection; we do not know if the sample size is adequate, as there is no sample size calculation; there are few women, the results may not be generalised to the female population; most of the outcomes measures are based on self-report questionnaires, meaning some participants may have under- or over-reported their SMD, Sensation Seeking, or SUD for a variety of reasons (e.g., difficulty in recalling quantity and frequency of SU).
References
- Only 27 articles out of 75 are published in the last five years. Please, update your references and remove unnecessary ones.
- Review the bibliography guidelines according to the journal's standards.
- Reference 25 has a different format.
Reviewer 2 Report
- The influences of substance use on ADHD symptoms, sensation seeking and sensory modulation dysfunction should be considered. It is highly possible that ADHD symptoms, sensation seeking and sensory modulation dysfunction were the results of substance use.
- This study examined ADHD-related symptoms by MOXO-CPT but no the diagnosis of ADHD. The result of the Adult ADHD Self-Report Scale also indicated a high tendency for ADHD only. It may be better to revise the title into “Attention-deficit/hyperactivity disorder symptoms”.
- This study surveyed sensation seeking by the Brief Sensation Seeking Scale (BSSS). High scores on the four components of the BSSS (Experience seeking; Boredom susceptibility; Thrill and adventure seeking; and Disinhibition) may increase the possibility but do not have to lead to risk-taking behaviors. Therefore, the aims should be revised and used sensation seeking to replace “risk-related behaviors.”
- All ADHD symptoms, sensation seeking and sensory modulation dysfunction might high correlate with each other. The possible problem of multicollinearity should be examined.
- The influences of types and durations of substances used among the study group should be analyzed and discussed.
- “Our findings contribute significance for the better understanding of SUD, as well as suggest a new perspective for both prevention and rehabilitation as part of the broader SUD treatment.” The authors should explain what kinds of prevention and rehabilitation can be built base don the results of this study. In fact, there have been several intervention programs focusing on the neurocognitive training for substance users.
- Error: (95%CI = 5.965, 121.216; p=<.0001(.
- Error: Sensory modulation dysfunction (SMD)
Round 2
Reviewer 2 Report
The authors have revised their manuscript based on the reviewer's suggestions. Only one minor error warrants revision: Line 51 "Sensory modulation dysfunction (SMD)" should be "sensory modulation dysfunction (SMD)." I would like to suggest the editors accepting it for publication.